# Development and evaluation of loop-mediated isothermal amplification for detection of *Yersinia pestis* in plague biological samples

**Lovasoa N. Randriantseheno**[1,2], **Anjanirina Rahantamalala**[3], **Ando L. Randrianierenana**[2], **Minoarisoa Rajerison**[1], **Voahangy Andrianaivoarimanana**[1] *

1 Plague Unit, Institut Pasteur de Madagascar, Antananarivo, Madagascar, 2 Department of Applied and Fundamental Biochemistry, University of Antananarivo, Antananarivo, Madagascar, 3 Immunology of Infectious Diseases Unit, Institut Pasteur de Madagascar, Antananarivo, Madagascar

* kekely@pasteur.mg

## Abstract

### Background

Several tests are available for plague confirmation but bacteriological culture with *Yersinia pestis* strain isolation remains the gold standard according to the World Health Organization. However, this is a time consuming procedure; requiring specific devices and well-qualified staff. In addition, strain isolation is challenging if antibiotics have been administered prior to sampling. Here, we developed a loop-mediated isothermal amplification (LAMP) technique, a rapid, simple, sensitive and specific technique that would be able to detect *Y. pestis* in human biological samples.

### Methods

LAMP primers were designed to target the *caf1* gene which is specific to *Y. pestis*. The detection limit was determined by testing 10-fold serial dilution of *Y. pestis* DNA. Cross-reactivity was tested using DNA extracts from 14 pathogens and 47 residual samples from patients suffering from non-plague diseases. Specificity and sensitivity of the LAMP *caf1* were assessed on DNA extracts of 160 human biological samples. Then, the performance of the LAMP *caf1* assay was compared to conventional PCR and bacteriological culture.

### Results

The detection limit of the developed *Y. pestis* LAMP assay was 3.79 pg/μl, similar to conventional PCR. The result could be read out within 45 min and as early as 35 minutes in presence of loop primer, using a simple water bath at 63°C. This is superior to culture with respect to time (requires up to 10 days) and simplicity of equipment compared to PCR. Furthermore, no cross-reactivity was found when tested on DNA extracts from other pathogens and human biological samples from patients with non-plague diseases. Compared to the

**Data Availability Statement:** All relevant data are within the manuscript and its Supporting Information files.

**Funding:** This work was supported by the Institut Pasteur de Madagascar. The funders had no role in study design, data collection and analysis, decision to publish, or preparation of the manuscript.

**Competing interests:** The authors have declared that no competing interests exist.

gold standard, LAMP sensitivity and specificity were 97.9% (95% CI: 89.1%-99.9%) and 94.6% (95% CI: 88.6%-97.9%), respectively.

## Conclusion

LAMP detected *Y. pestis* effectively with high sensitivity and specificity in human plague biological samples. It can potentially be used in the field during outbreaks in resource limited countries such as Madagascar.

## Introduction

Plague is a zoonotic disease caused by a gram-negative bacterium, *Yersinia pestis*, which has been responsible for three major historical pandemics leading to millions of deaths. Currently, plague is endemic in some American, Asian and African countries including Madagascar which reports the vast majority of human plague cases across the globe (85.93% of global cases in 2015) [1]. The disease remains in foci that are mainly located in countries of extreme poverty. The World Health Organization (WHO) has classified plague as a re-emerging infectious disease. The recent pneumonic plague outbreak occurring in 2017 in two densely populated urban areas in Madagascar [2] reminds us that plague is far from being eradicated and its control is still challenging. Moreover, most of human cases originate and surface in rural low resourced regions where local health centers suffers from very limited diagnostic capability to robustly identify plague.

Plague is a rapidly progressing disease that has two main clinical forms depending on the transmission route. Bubonic plague (BP), the most common form, is acquired following the bite of infected rodent fleas. From the inoculation site, *Y. pestis* reaches the lymph nodes where massive multiplication occurs. Pneumonic plague (PP), a rare but even more deadly form, can evolve from a BP complication or resulting from a human to human transmission. The 2017 PP outbreak in Madagascar [2] illustrates the dangers of undiagnosed plague and the urgent need for wide distribution of diagnostic capabilities throughout Madagascar, especially in low resource setting of rural health facilities.

In Madagascar, national law mandates that all suspected plague cases must be diagnostically confirmed and reported. Multiple types of diagnostic tools are employed in this effort; the easy-to-use rapid diagnostic test for F1 antigen detection (F1 RDT) [3], quantitative real-time PCR (qPCR) detecting two *Y. pestis* targets (*caf1* and *pla* genes) [2], an anti-F1 antibody ELISA [4] and the bacteriological culture with *Y. pestis* strain isolation [5]. This last method remains the gold standard for plague confirmation according to the WHO [6]. However, culture is time-consuming (10 days minimum) and successful strain isolation is highly dependent on other variables such as quality of sample conservation and low amount of contaminating bacteria. In addition, success of isolation is lower in samples from patients who started their treatment with antibiotics. A conventional PCR targeting the *caf1* gene was previously developed and assessed but was not recommended as a routine diagnostic test for plague in Madagascar due to its low sensitivity compared to culture [7]. Although RDT and qPCR/conventional PCR are of much value in facilitating major improvements in disease management, when each are taken alone they do not constitute a confirmatory test. A confirmatory result is constituted through either a combination of tests (positive RDT and qPCR/conventional PCR) or an ELISA test result showing four-fold rise in anti-F1 antibody titer in paired-sera [6].

Official confirmatory testing is conducted at the Central Laboratory for Plague (CLP) hosted at the Plague Unit of the Institut Pasteur de Madagascar (IPM) located in the capital Antananarivo. The main reason these tests are conducted at IPM and not in local rural hospitals is because most require complex technologies and technical expertise which are not available in rural hospitals (except for the rapid test F1 RDT). The lack of diagnostic capability in rural hospitals likely contributes to the underreporting of plague. The delays and logistical burdens of transporting biological specimens to IPM can hamper official confirmation by resulting in unsuccessful strain isolation (gold standard confirmation). The rapidness of disease progression and the danger of underreporting disease illustrate the urgent need for robust diagnostic capabilities in low resource rural health facilities.

A diagnostic technology useable in a low resource setting would need to be simple, rapid, specific and cost-effective. Loop-mediated isothermal amplification (LAMP) is a rapid, efficient and specific DNA amplification method developed in 2000 by Notomi *et al* [8]. LAMP technique has been used for various infectious diseases diagnosis: visceral leishmaniasis [9], tuberculosis [10], malaria [11], bacillary dysentery [12], rickettsiosis [13], african trypanosomiasis [14], Zika virus [15]. LAMP methods detecting *Y. pestis* have previously been developed but evaluated only on *Y. pestis* pure cultures [16] or on simulated samples [17].

This technique relies on autocycling strand displacement coupled to DNA synthesis by *Bst* DNA polymerase. This eliminates the necessity for the heat denaturation step. The use of a set of four to six primers (two outer primers F3 and B3, two inner primers FIP and BIP, and two optional loop primers LF and LB) are responsible of its great sensitivity and specificity. The simplicity and the rapidness of the technique are associated with the isothermal condition of the reaction carried out for approximately 1 hour at 60°C to 65°C (requiring a simple water bath or heating block) and the results can be read by the naked eye with visual turbidity [18] or visual fluorescence (calcein [19], propidium iodide [20]) or using colorimetric agents (such as hydroxynaphtol blue [21]) or intercalating agent (such as SYBR green I [8]).

In this study, we developed a rapid, simple and sensitive/specific LAMP method assay for the detection of the *caf1* gene sequence that is specific to *Y. pestis* and to evaluate its performance on biological samples from plague suspected patients from Madagascar.

## Materials and methods

### Ethics statement

The DNAs used in this study were extracted from *Y. pestis* cultures or human biological samples originally isolated or collected by the CLP and IPM as part of the plague national control program (PNCP) of the Malagasy Ministry of Public Health (S1 Appendix). This PNCP requires declaration of all suspected human plague cases and collection of biological samples from those cases. These samples and any cultures or DNA derived from those samples were collected under this mandatory reporting system and thus exempt as human subjects research. All culture isolates and biological samples were de-linked from the patients' identifiable information and analyzed anonymously. Therefore no approval from the Malagasy Ethical Committee was required for this study.

### *Y. pestis* strains and human biological samples

*Y. pestis* strains used in this study were isolated in 2015. A total of 113 biological samples from suspected plague patients (88 bubo aspirates, 12 sputum samples, 13 post-mortem samples consisting of 8 samples of lung punctures and 5 of liver punctures) stored at the CLP-Plague Unit Collection were used to evaluate the LAMP *caf1* technique. Forty-nine of the biological samples were bacteriology positive with strain isolation and the remaining 64 were negative

for all tests available (i.e. F1 RDT, qPCR and bacteriology). For ethical reasons, it was not possible to obtain negative control bubo aspirates or sputum samples from healthy populations. Therefore, as negative controls, we used 47 samples (29 pus and 18 sputum) from patients confirmed with other infectious diseases not consistent with plague.

### DNA extraction

Bacterial strains were sub-cultured in brain-heart infusion broth at 26˚C for 48h. The culture (1.5 ml) was then subjected to DNA extraction using DNeasy Blood & Tissue Kit (Qiagen, Germany) according to the manufacturer's protocols for "Pretreatment for Gram-Negative Bacteria" and "Purification of Total DNA from Animal Tissues (Spin-Column Protocol)" [22].

DNA extracts from biological samples were obtained using a method previously described by Rivoarilala et al. [23] with slight modification. Samples were placed into a boiling water bath for 10 min and centrifuged at 12,000 rpm for 5 min. Five microliters (5 μl) of the supernatant were used for LAMP assays.

### LAMP primer design

The *caf1* sequence previously targeted by Rahalison *et al.* [7] for conventional PCR was recovered and used as a template to design LAMP primers. Primers F3, B3, FIP and BIP were automatically designed using the Primer Explorer V.4 software (https://primerexplorer.jp/elamp4.0.0/). The primers sequences selected for our LAMP system and used on the biological samples in this study are shown in Table 1.

### Optimization of LAMP amplification protocol

The fully validated LAMP reaction mixture contained the following reagents for a final reaction volume of 25 μl: 1X thermopol buffer (New England Biolabs), 0.95 M of betaine (Sigma-Aldrich), 7 mM of $MgSO_4$ (New England Biolabs), 1.4 mM of deoxynucleoside triphosphate (dNTPs) (Invitrogen), 0.13 μM each of F3 and B3 primers (Sigma-Aldrich), 1.06 μM each of FIP and BIP primers (Sigma-Aldrich), 8U *Bst* DNA polymerase (New England Biolabs) and 5 μl of positive control DNA (*Y. pestis* strain 59/15) or negative control (sterile distilled water). To identify the conditions optimal for maximal amplification, the following parameters were tested within a specified range: reaction times (30–60 min), incubation temperatures (57–69˚C), and betaine concentrations (0 and 0.95 M). With the goal of employing simpler equipment, we tested incubations of LAMP reaction mixture on a Veriti 96 Well Thermal Cycler (Applied Biosystems) in parallel with a water bath to confirm that the bath worked equally well to that of the thermal cycler. All experiments testing the parameters included duplicate samples and were repeated twice to ensure repeatability.

**Table 1. Nucleotide sequences of the set of LAMP *caf1* primers designed in this study.**

| Primer | Sequence (5'-3') |
|---|---|
| F3 | CGGGTGATCCCATGTACT |
| B3 | CATCAGTGTATTTACCTGCTG |
| FIP | ATCAAAATCTCTAGAATCCTTGCCATTTTCTCAGGATGGAAATAACCACC |
| BIP | GGATGACGTCGTCTTGGCTATTTTCAAGTTTACCGCCTTTGG |

F3: forward outer primer, B3: backward outer primer, FIP: forward inner primer, BIP: backward inner primer

## Analysis of LAMP products

To determine if the LAMP reaction amplified DNA, 2 μl of 10-fold diluted SYBR Green I 10,000X (Sigma-Aldrich) was added directly into the tube containing the LAMP reaction mix. SYBR Green I, when bound to double stranded DNA, emits green light which can be visualized by naked eye. A positive LAMP amplification is indicated by a color change from orange to green in the reaction tube whereas no amplification is indicated by the color remaining orange (no color change) (Fig 1). To independently confirm positive DNA amplification, 5 μl of LAMP reaction mix were visualized by electrophoresis using 1.5% agarose gel stained with ethidium bromide. Characteristic ladder-like bands of multiple sizes and no band were shown for positive and negative reactions, respectively [8].

## Loop primers to enhance LAMP amplification

The use of loop primers is not compulsory but was reported to reduce LAMP reaction time and improve specificity [24]. Therefore, four backward loop primers (LB) were designed with Primer Explorer V.4 (Table 2) and tested at 0.38 μM.

## Outer primers (F3 and B3) specificity

Conventional PCR using the outer primers (F3 and B3 LAMP primers) was performed in order to confirm the specificity of the targeted region amplification. The sequence of *Y. pestis* CO92 plasmid pMT1 (GenBank Accession No. AL 117211.1) was used to localize the primer

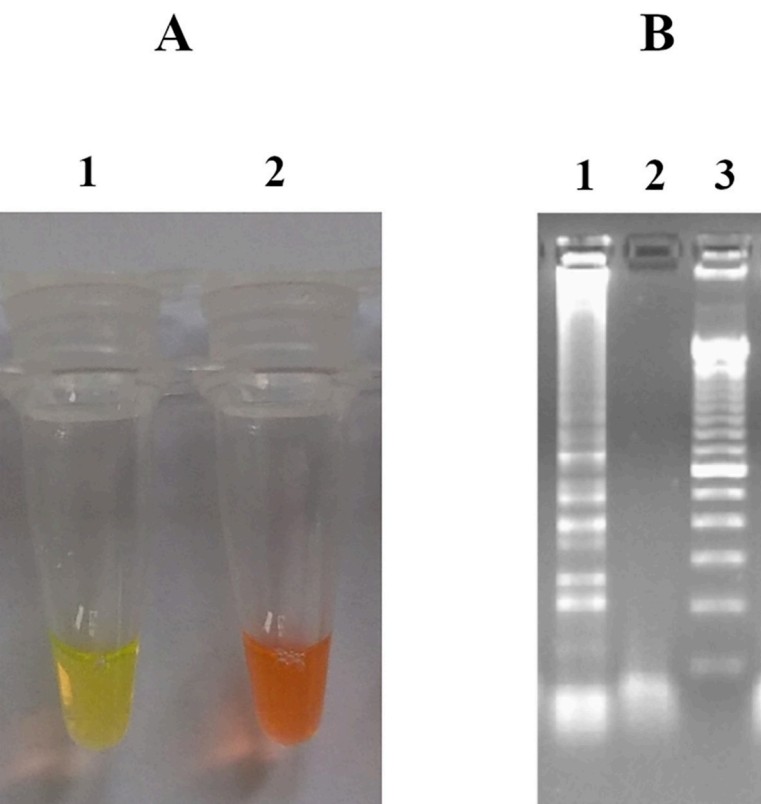

**Fig 1. Visualization of LAMP products.** (A) stained with SYBR Green I and observed under natural light (Tube 1: *Y. pestis*, Tube 2: negative control); (B) with agarose gel electrophoresis (Lane 1: *Y. pestis*, Lane 2: negative control, Lane 3: DNA ladder marker 100 bp).

**Table 2. Sequences of backward loop primers.**

| Name | Sequence (5'-3') |
| --- | --- |
| LB1 | `CAGCCAGGATTTCTTTGTTCGCTCA` |
| LB2 | `AGCCAGGATTTCTTTGTTCGCTCA` |
| LB3 | `GCCAGGATTTCTTTGTTCGCTCA` |
| LB4 | `CCAGGATTTCTTTGTTCGCTCA` |

position on the targeted region with Sequence Extractor (https://www.bioinformatics.org/seqext/). The size of the targeted region was determined and compared to the size of the PCR amplicons obtained with F3 and B3. The conventional PCR reaction volume was 25 µl and contained 1X of CoralLoad® PCR Buffer (Qiagen), 0.4 mM of dNTPs (Invitrogen), 0.06 mM each of F3 and B3 primers (Sigma-Aldrich) (Table 1), 1U of Taq DNA Polymerase (Qiagen) and 5 µl of *Y. pestis* DNA extract for positive control and sterile distilled water for negative control. The test of specificity was carried out on seven different DNA extracts from *Y. pestis* strains 39/15, 56/15, 59/15, 69/15, 70/15, 72/15 and 73/15. The reaction comprised an initial denaturation step of 3 min at 94˚C, 40 cycles of 94˚C for 30 sec, 58˚C for 30 sec, 72˚C for 30 sec and a final extension of 10 min at 72˚C. Amplicons were analyzed on 1.5% agarose gel stained with ethidium bromide during 60 min.

## Detection limit of LAMP and conventional PCR

To characterize the minimum concentration of *Y. pestis* DNA detectable by LAMP *caf1* and conventional PCR, *Y. pestis* DNA (initial concentration: 37.9 ng/µl) was 10-fold serially diluted with sterile distilled water, ranging from $10^{-1}$ to $10^{-5}$ and tested in parallel with LAMP and conventional PCR. The performance of LAMP *caf1* was compared to that of the conventional PCR using primers previously tested by Rahalison *et al.*, (Forward 5' – `CAGTTCCGTTAT CGCCATTGC`– 3' and reverse 5' – `TATTGGTTAGATACGGTTACGGT`– 3', with 501 bp of expected amplification product) [7]. The reaction mixture and the program were the same as for specificity assessment of the outer primers described above.

## Cross-reactions

Cross-reactions with other *Yersinia spp*, other bacterial or parasitological diseases prevalent in Madagascar were assessed. Five microliters (5 µl) of DNA extract of each strain (*Yersinia enterocolitica*, *Yersinia pseudotuberculosis*, *Escherichia coli*, *Enterobacter cloacae*, *Shigella sonnei*, *Proteus mirabilis*, *Serratia odorifera*, *Serratia marcescens*, *Pseudomonas aeruginosa*, *Staphylococcus aureus*, *Taenia solium*, *Plasmodium vivax*, *Plasmodium falciparum* and 4 of *Mycobacterium tuberculosis*) was tested. Concentrations of all DNA extracts were measured with spectrophotometer (Nanodrop 2000). Forty-seven biological samples tested positive with other diseases (schistosomiasis (n = 7), taeniasis (n = 3), cysticercosis (n = 8), filariosis (n = 1) and tuberculosis (n = 28)) were also tested to better assess potential cross-reactions.

## Statistical analysis

LAMP *caf1* was evaluated with the pre-treated human biological samples using the determined optimal conditions. Results were compared to those of the reference method (bacteriological culture). Specificity and sensitivity of LAMP *caf1* were calculated with 95% confidence intervals using R 3.6.2. The kappa coefficient (κ) was also calculated to assess the level of agreement between the index test (LAMP *caf1*) and the reference test (bacteriological culture).

# Results

## Optimization of LAMP reaction

The optimal LAMP reaction was obtained using 0.95 mM betaine at 63˚C after 45 min of amplification (Fig 2). Similar results were obtained using a thermal cycler and a simple water bath in parallel. Therefore, subsequent LAMP assays tests were conducted using water bath.

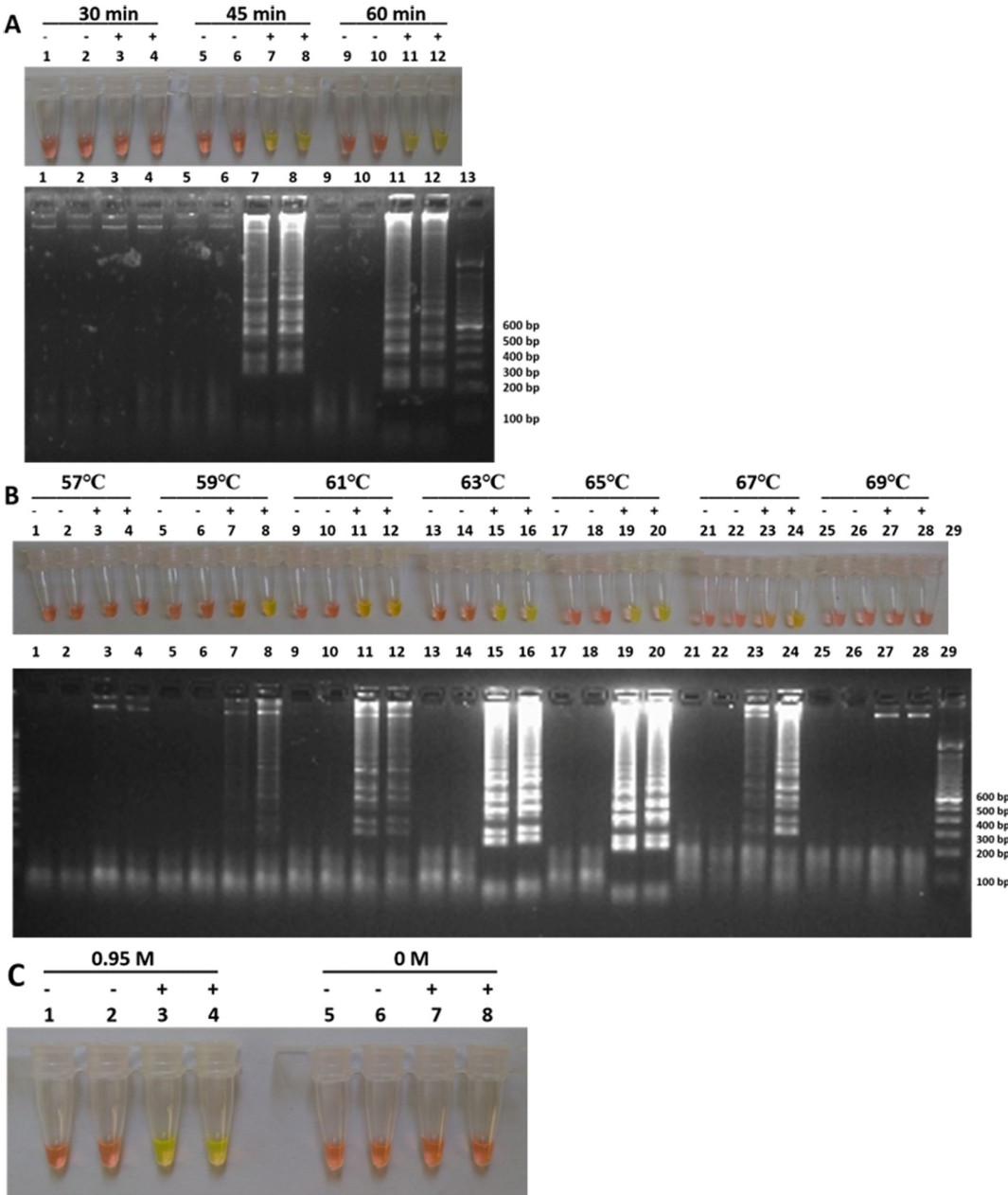

**Fig 2. Optimization of LAMP *caf1*.** Optimal reaction was found by varying LAMP reaction parameters. Amplification was verified by naked eye with color change from orange to green (positive test) and without color change remaining orange (negative test) and by agarose gel electrophoresis. Optimization results of (A) the reaction time: 30 min (Tubes and Lanes 1–4), 45 min (Tubes and Lanes 5–8) and 60 min (Tubes and Lanes 9–12) (B) the reaction temperature: 57˚C (Tubes and Lanes 1–4), 59˚C (Tubes and Lanes 5–8), 61˚C (Tubes and Lanes 9–12), 63˚C (Tubes and Lanes 13–16), 65˚C (Tubes and Lanes 17–20), 67˚C (Tubes and Lanes 21–24) and 69˚C (Tubes and Lanes 25–28) (C) the betaine concentration: 0 M (Tubes 1–4) and 0.95 M (Tubes 5–8).

When we tested the four LB (Table 2), they were all effective for target sequence amplification. Therefore, LB1 was kept for LAMP *caf1* assays, making this a five primer *Y. pestis* LAMP assay. The target fragment was well amplified using LB1 after 35 min.

## Specificity of the outer primers (F3 and B3)

The positions of the outer primers (F3 and B3) on the *Y. pestis* CO92 plasmid pMT1 (GenBank Accession No. AL 117211.1) determined with Sequence Extractor is shown in S2 Appendix. The expected target sequence was found to be 225 bp long (located between 86209–86433) which corresponded to the length of the obtained PCR amplicons (Fig 3). The conventional PCR specificity test did not show the presence of other additional bands using 7 DNA extracts confirming the specificity of the outer primers used for LAMP assay.

## Detection limit of LAMP and PCR assays

When amplifying serial dilution templates of *Y. pestis* DNA, the detection limit of LAMP *caf1* was found to be 3.79 pg/μl ($10^{-4}$) (Fig 4). The detection limit of LAMP *caf1* was similar to that of PCR *caf1*.

## Cross-reactions

When amplifying DNA extracts from 14 other pathogens, a false positive result was observed with one DNA extract out of the four tested for *M. tuberculosis* (1) after SYBR Green I addition (Fig 5A). Further confirmation of the amplification products by electrophoresis showed no ladder-like pattern characteristic of LAMP products but a smear revealing the absence of LAMP amplification (Fig 5B). We hypothesis that this smear was pre-existing and not generated by the LAMP amplification. High specificity of LAMP *caf1* was indicated by the lack of cross-reaction when the assay was tested against 14 pathogens and 47 plague-negative biological samples.

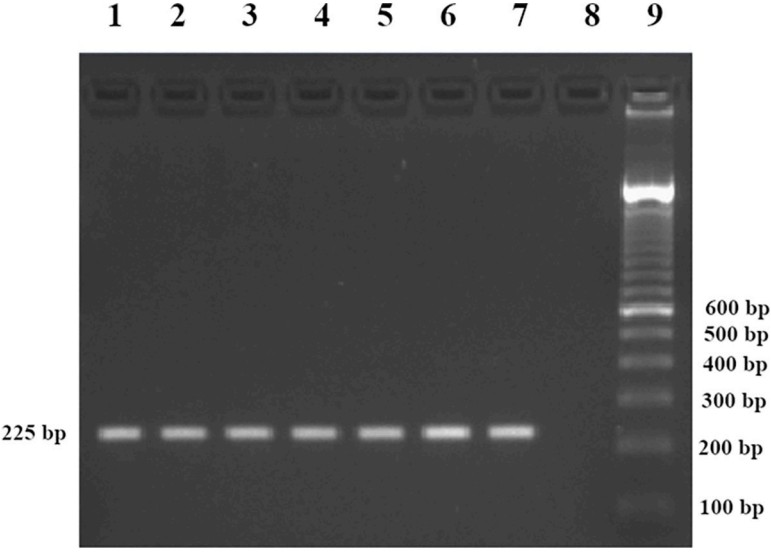

**Fig 3. PCR amplifying the target sequence in 7 *Y. pestis* strains with the outer primers (F3 and B3).** Lanes 1 to 7: *Y. pestis* DNA; Lane 8: no DNA template (sterile distilled water); Lane 9: DNA ladder marker 100 bp.

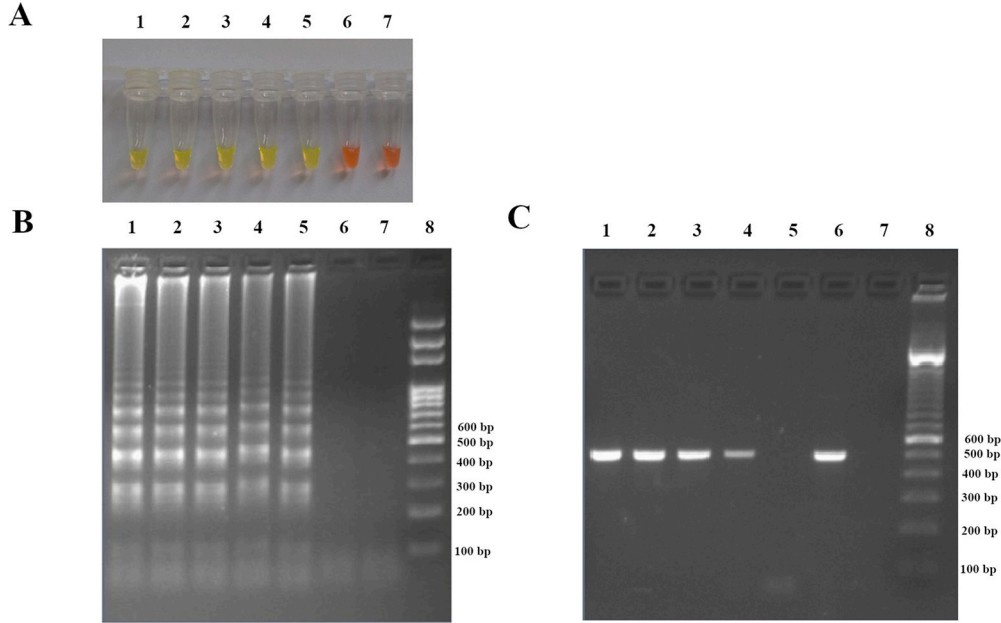

**Fig 4. Detection limit of LAMP and PCR *caf1*.** Ten-fold serial dilutions of *Y. pestis* DNA extract were tested. (A) Visualization of color change by the naked eye. (B) Confirmation of results by agarose gel electrophoresis. Tubes and Lanes 1, 2, 3, 4, 5, 6, and 7: *Y. pestis* DNA extracts undiluted stock, $10^{-1}$, $10^{-2}$, $10^{-3}$, $10^{-4}$, $10^{-5}$, and no DNA template (sterile distilled water) respectively, Lane 8: DNA ladder marker 100 bp. (C) Detection limit of conventional PCR *caf1*. Lanes 1, 2, 3, 4, 5, 6, and 7: $10^{-1}$, $10^{-2}$, $10^{-3}$, $10^{-4}$, $10^{-5}$, *Y. pestis* DNA extracts undiluted stock and no DNA template (sterile distilled water) respectively, Lane 8: DNA ladder marker 100 bp.

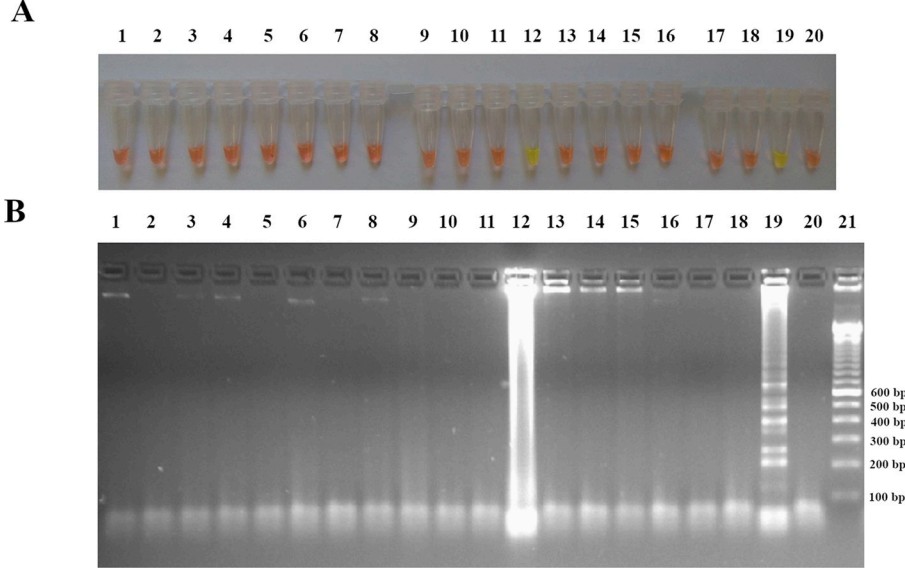

**Fig 5. Cross-reactions of LAMP *caf1*.** LAMP reaction was assessed for DNA amplification of 14 pathogens: (A) Eye visualization of LAMP reaction after SYBR Green I addition, (B) visualization after gel electrophoresis migration of the LAMP products. Tubes and Lanes 1: *Y. enterocolitica*, 2: extraction control, 3: *Y. pseudotuberculosis*, 4: *E. cloacae*, 5: *E. coli*, 6: *S. sonnei*, 7: *P. mirabilis*, 8: *S. odorifera*, 9: *S. marcescens*, 10: *P. aeruginosa*, 11: *S. aureus*, 12: *M. tuberculosis* (1), 13: *M. tuberculosis* (2), 14: *M. tuberculosis* (3), 15: *M. tuberculosis* (4), 16: *P. vivax*, 17: *P. falciparum*, 18: *T. solium*,19: *Y. pestis*, 20: no DNA template (sterile distilled water) and Lane 21: DNA ladder marker 100 bp.

**Table 3. Evaluation of LAMP *caf1* performance.**

| LAMP *caf1* | Bacteriological culture | | |
|---|---|---|---|
| | Positive | Negative | Total |
| Positive | 48 | 6 | 54 |
| Negative | 1 | 105 | 106 |
| Total | 49 | 111 | 160 |

## Sensitivity and specificity of LAMP *caf1*

A total of 160 human biological samples were tested for LAMP *caf1* evaluation. Of the 49 plague culture positive samples tested, 47 were found positive, 1 negative and 1 inconclusive with LAMP *caf1* based on naked eye observation with SYBR green I. The inconclusive LAMP reaction was from a sample that was dark in color before LAMP amplification. This dark coloration of the sample interfered with the ability to detect color change with Sybr Green I. However, visualization on 1.5% agarose gel electrophoresis showed faint LAMP ladder-like bands pattern. The faint DNA signal combined with the ladder-like bands after amplification seemed to suggest low concentration of the starting template DNA. The sensitivity of the LAMP assay compared to the bacteriological culture was 97.9% (95% CI: 89.1%-99.9%). Of the 111 culture negative samples, 105 remained negative with LAMP *caf1* resulting to a specificity of 94.6% (95% CI: 88.6%-97.9%) (Table 3). A kappa coefficient of 0.89 (95% CI: 0.83%-0.97%) was obtained.

## Discussion

Rapid diagnosis at the very early acute phase of plague is key to preventing mortality and spread through human to human transmission. Since plague outbreak mainly occurs in rural remote regions, a highly simple, inexpensive and rapid diagnosis would need to occur locally. The F1 RDT is widely used at the national level but, alone, is not considered an official confirmation. The current WHO gold standard for plague confirmation is the isolation of *Y. pestis* strains from biological samples [6], a time-intensive procedure that can take up to 10 days for results and requires a special biosecurity infrastructure facility that is non-existent in low-resource settings. Since the 2017 pneumonic plague outbreak in Madagascar, the CLP of the Malagasy Ministry of Public Health adopted a new diagnostic scheme for plague cases confirmation that includes, in addition to bacteriological culture, the combination of F1 RDT and duplex qPCR assays (detecting *caf1* and *pla* genes) [2]. Although qPCR is rapid, yielding results in less than 4 hours, it requires sophisticated expensive equipment that is cost-prohibitive for regional hospital laboratories in developing countries. In this study, we described a new LAMP technique with high sensitivity and specificity for the rapid detection of *Y. pestis* and only require low-cost equipment. Combined with F1 RDT, LAMP *caf1* assay would endow confirmatory capabilities at the local health clinics level throughout developing countries where plague foci are still endemic.

LAMP *caf1* is significantly more rapid than qPCR, with 50 min (15 min "boil & spin" and 35 min amplification) vs less than 4 hours of completion time. The duplex qPCR also requires extraction of DNA from samples which places added resource burden. Two LAMP assays detecting *Y. pestis* have already been described targeting *caf1* gene [16] and the 3a sequence on *Y. pestis* chromosome [17]. Both assays were evaluated on *Y. pestis* strains but not on infected biological samples. To the best of our knowledge, our study is the first to demonstrate the successful diagnostic use of the LAMP method on *Y. pestis* infected biological samples. In addition to the DNA extraction step, previously published *Y. pestis* LAMP assays required longer

amplification time than our assay by 10 to 25 minutes (45 min [16] and 60 min [17]) and an additional 5 min step of either denaturation [16] or inactivation [17].

Our results showed a similar detection limit (3.79 pg/μl) between LAMP *caf1* assay and conventional PCR *caf1* using *Y. pestis* DNA. Although a similar result was previously reported with *Y. pestis* LAMP assay [16], several studies characterizing LAMP techniques on other pathogens reported lower detection limit compared to conventional PCR [25–29]. However, LAMP *caf1*, using five different primers, has the advantage of being more robust compared to conventional PCR (with 2 primers). The LAMP assay was reported to be tolerant to certain types of biological substances that are not well tolerated in conventional PCR [30] thus making the additional steps to clean DNA template not necessary. A simple, rapid and cheaper ("boil & spin") pre-treatment of the biological samples is enough for LAMP technique and makes it a significantly more feasible tool in low resource settings.

Compared to bacteriological culture, our LAMP *caf1* showed efficacy for the detection of *Y. pestis* in biological samples with a sensitivity of 97.9% (95% CI: 89.1%-99.9%) and a specificity of 94.6% (95% CI: 88.6%-97.9%). The kappa coefficient of 0.89 (95% CI: 0.83%-0.97%) reflects an almost perfect agreement between bacteriological culture and LAMP *caf1*.

Certain parameters influenced LAMP *caf1* performance. With regard to the reagents, it has been hypothesized that only GC rich sequences require the use of betaine [25]. In our case, betaine was found to be compulsory for LAMP *caf1* even though the %GC of the target sequence was only 44.4%.

The duration of the reaction decreased by 10 min when adding the backward loop primer (45 min without LB1 vs 35 min). Other studies already emphasized similar findings and explained that when the loop primer is used, the initiation of synthesis at several regions resulted in a drastic amplification of the target sequence [24].

Although LAMP *caf1* assay is highly promising, the technique presents some limitation and some challenges. This technique would not be able to detect *Y. pestis* strains with a deletion of all or part of the *caf1* gene or a loss of the entire pFra/pMT1 plasmid [7, 31] but these modifications are rare for *Y. pestis*. Our LAMP assay uses SYBR Green I to visually confirm amplification of DNA. SYBR Green I proves not to be compatible when template DNA concentrations are high or the biological samples are dark in coloration. Firstly, when assessing LAMP *caf1* for cross-reaction, 1 out of 4 *M. tuberculosis* DNA samples showed positive result after SYBR Green I addition (Fig 5). SYBR Green I is an intercalating dye binding non-specifically to all double stranded DNA, thus a sample containing high concentration of the DNA template would give a positive result without amplification of the target sequence. The resulting quantification of the *M. tuberculosis* DNA concentration (1171.9 ng/μl) using spectrophotometer seems to support this hypothesis. To overcome this challenge, DNA concentration can be quantified before the start of LAMP amplification to avoid the case of false positive result using SYBR Green I. If no DNA quantification method is available, another colorimetric method such as addition of hydroxy naphtol blue [21] or malachite green [32] may be more appropriate. An alternative approach used in this study was the addition of SYBR Green I to a representative aliquot of DNA before amplification. A green color indicates that the DNA extract is too concentrated and should be diluted before starting the reaction. We did not encounter false positive result once this *M. tuberculosis* (1) DNA extract was diluted before amplification. Overall, no cross-reaction was found against the remaining pathogens tested and on negative human samples thus demonstrating the high specificity of the amplification with LAMP *caf1*. Secondly, the use of colorimetric agent such as SYBR Green I on dark colored biological samples was incompatible due to interfering with the ability to visualize the color change. In our case, after gel electrophoresis of a dark colored sample, we found it to be positive with blurred ladder-like bands pattern indicating a low concentration of template DNA.

Therefore, we concluded that analyzing amplified products using colorimetric methods are not suitable for dark colored samples, especially if samples had low DNA concentration. These findings are in agreement with a previous publication [33]. Each challenge encountered had alternative ways of resolving which may not always be suitable for point-of-care use.

In conclusion, LAMP *caf1* developed in this study detected *Y. pestis* effectively in human biological samples with remarkable levels of sensitivity (97.9%) and specificity (94.6%). The use of this technique will save considerable time and effort which is particularly important for a fatal disease like plague that progresses rapidly without treatment. If combined with F1 RDT, its use would make it suitable as a potential confirmation method within plague endemic countries. Although further evaluation is needed in rural remote settings and training should be provided to healthcare staff who are not familiar to molecular technique, this LAMP assay holds great promise due to its simplicity and high performance. Adoption of this assay would greatly help address the urgent need to endow robust diagnostic capabilities at the local level in plague endemic foci which are mainly located in rural remote regions throughout Madagascar.

## Supporting information

**S1 Appendix. Plague national control program.**
(PDF)

**S2 Appendix. Delimitation of the sequence targeted by the outer primers (F3 and B3) using sequence extractor.**
(TIF)

**S1 Raw images. Original images supporting Fig 1(B), Fig 2(A), Fig 2(B), Fig 3, Fig 4(B), Fig 4(C) and Fig 5(B).**
(PDF)

## Acknowledgments

We would like to thank Pr Milijaona Randrianarivelojosia (Malaria Research Unit, Institut Pasteur de Madagascar) for the kind provision of DNA extracts of *Plasmodium spp*. We also thank Dr Niaina Rakotosamimanana (Mycobacteria Unit, Institut Pasteur de Madagascar) and Dr Frédérique Randrianirina (Clinical Center of Biology, Institut Pasteur de Madagascar) for the kind provision of DNA extracts of *M. tuberculosis* and residual samples from unsuspected plague patients used for the evaluation test in this study.

We are grateful to Dr Dawn Birdsell for editing this manuscript.

## Author Contributions

**Conceptualization:** Minoarisoa Rajerison, Voahangy Andrianaivoarimanana.

**Data curation:** Lovasoa N. Randriantseheno.

**Formal analysis:** Lovasoa N. Randriantseheno, Anjanirina Rahantamalala, Voahangy Andrianaivoarimanana.

**Funding acquisition:** Minoarisoa Rajerison.

**Investigation:** Lovasoa N. Randriantseheno, Anjanirina Rahantamalala.

**Methodology:** Lovasoa N. Randriantseheno, Anjanirina Rahantamalala, Ando L. Randrianierenana, Voahangy Andrianaivoarimanana.

**Project administration:** Minoarisoa Rajerison, Voahangy Andrianaivoarimanana.

**Software:** Lovasoa N. Randriantseheno, Voahangy Andrianaivoarimanana.

**Supervision:** Ando L. Randrianierenana, Minoarisoa Rajerison.

**Visualization:** Lovasoa N. Randriantseheno.

**Writing – original draft:** Lovasoa N. Randriantseheno.

**Writing – review & editing:** Anjanirina Rahantamalala, Ando L. Randrianierenana, Minoarisoa Rajerison, Voahangy Andrianaivoarimanana.

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
