## [Decision Letter · Decision Letter 0]

17 Apr 2020

PONE-D-20-06493

Development and evaluation of loop-mediated isothermal amplification for detection of Yersinia pestis in plague biological samples

PLOS ONE

Dear Dr. Andrianaivoarimanana,

Thank you for submitting your manuscript to PLOS ONE. After careful consideration, we feel that it has merit but does not fully meet PLOS ONE’s publication criteria as it currently stands. Therefore, we invite you to submit a revised version of the manuscript that addresses the points raised during the review process.

In particular your are asked to further study the diagnostic performance of the developed assay by incliuding more samples, in particular of healthy endemic controls and other diseases.

The employed statistics require further attention

The issues about diagnostic performance should be addressed in an inteligent manner

All issues and points raised by the reviewers must be addressed

The figures need to be of better quality

We would appreciate receiving your revised manuscript by Jun 01 2020 11:59PM. To enhance the reproducibility of your results, we recommend that if applicable you deposit your laboratory protocols in protocols.io, where a protocol can be assigned its own identifier (DOI) such that it can be cited independently in the future. For instructions see: http://journals.plos.org/plosone/s/submission-guidelines#loc-laboratory-protocols

We look forward to receiving your revised manuscript.

Kind regards,

Henk D. F. H. Schallig, Ph.D

Academic Editor

PLOS ONE

3. We note you have two different tables labeled as Table 3. Please ensure that all your tables are numbered separately.

Reviewers' comments:

Reviewer's Responses to Questions

**Comments to the Author**

1. Is the manuscript technically sound, and do the data support the conclusions?

Reviewer #1: Yes

Reviewer #2: Partly

2. Has the statistical analysis been performed appropriately and rigorously? 

Reviewer #1: I Don't Know

Reviewer #2: No

3. Have the authors made all data underlying the findings in their manuscript fully available?

Reviewer #1: Yes

Reviewer #2: Yes

4. Is the manuscript presented in an intelligible fashion and written in standard English?

Reviewer #1: Yes

Reviewer #2: Yes

5. Review Comments to the Author

Reviewer #1: The manuscript by Lovasoa Nomena Randriantseheno describes the development of a LAMP assay that would be able to detect Y. pestis in human biological samples, and thereby improving the diagnosis of plague and circumventing cumbersome bacteriological culture.

The authors claim that no ethical approval is needed fort his study and that there is no informed consent needed from study cases. Although I believe that this might be right in this specific case a document supporting this would be good to have.

The work aimed to develop a rapid, simple and sensitive/specific LAMP method for the detection of the caf1 gene sequence that is specific to Y. pestis and to evaluate its performance on biological samples from plague suspected patients from Madagascar. However, apparently previously a LAMP assay has been developed see line 100. The authors should specify why they undertook the current study. What is their novel approach.

The development of the LAMP assay is straightforward and well described.

Diagnostic characteristics of the test would have been more convincing if also patient samples of cases with other diseases were tested. Can the authors do this?

Sensitivity and specificity should be presented with 95% confidence intervals (also in the abstract). Agreement between tests (kappa values) must be estimated and presented. In particular there should be a clear definition of the reference test. This is now presented partly in the discussion but must be included in the Methods section and presented in the Results section.

Figures are not of very good quality. This must be improved.

Reviewer #2: The manuscript addresses the development of a loop-mediated isothermal amplification (LAMP) test to aid in the diagnosis of Plague by detecting Yersinia pestis nucleic acids in clinical samples.

The authors explain the advantages of polymerase chain reaction (PCR) in the diagnosis of Plague, in opposition to culture and antigen detection rapid diagnostic test (RDT). The aim of the study is to develop a LAMP method that being simpler than PCR can replace it and implemented at lower levels of the health system, provided it has good diagnostic performance.

The methodology used is appropriate, but I would like to discuss some aspects of this work:

1) According to the authors PCR alone is not enough to confirm a diagnosis of Plague and this requires also a positive RDT result or a 4X increase in antibody titre, why is this required? Please, explain. Why not aiming at a LAMP method that can be used alone as confirmatory tool?

2) The LAMP method proposed is home brewed, will this be accepted by the Ministry of Health? Is the PCR method also home brewed?

3) When cross-reactions were assessed was DNA preparation and measurement the same for all samples? This is not clear.

4) LAMP shows lower sensitivity than culture. Is not this a suboptimal performance? Also, the use of SYBRGreen is a big limitation, the authors suggest to quantify DNA first and adjust DNA concentration, which is not ideal for a point-of-care (POC) test. Also, for coloured DNA samples analysis of amplified products by PCR is also suggested, which again is far from a POC.

5) Sensitivity and specificity are not estimated with confidence intervals, please apply the statistics needed for this.

6) The resulting LAMP test, as it is, does not seem to bring any advantage to the diagnosis of Plague

6. PLOS authors have the option to publish the peer review history of their article (what does this mean?). If published, this will include your full peer review and any attached files.

Reviewer #1: No

Reviewer #2: No

---

## [Author Response · Author response to Decision Letter 0]

27 May 2020

Response to Reviewers

PONE-D-20-06493

Development and evaluation of loop-mediated isothermal amplification for detection of Yersinia pestis in plague biological samples

PLOS ONE

PLoS ONE Academic editor requirements

In particular you are asked to further study the diagnostic performance of the developed assay by incliuding more samples, in particular of healthy endemic controls and other diseases.

Answer: For ethical reasons, it was not possible to obtain negative control adenitis aspirates from patients without suspected plague. Therefore, samples from negative patients tested on our LAMP caf1 were constituted of samples from patients not suspected for plague which were archived at the Clinical Center of Biology (Institut Pasteur de Madagascar) for other analysis: sputum (n= 18) and pus (n= 29)

We were also able to add 47 patient samples with other diseases to test on our LAMP caf1. They are distributed as follows: schistosomiasis (n= 7), taeniasis (n= 3), cysticercosis (n= 8), filariosis (n= 1) and tuberculosis (n= 28). They were kindly provided by other research units from the Institut Pasteur de Madagascar from their samples collection.

The employed statistics require further attention

Answer: Thank you for pointing this out. Estimations of the Sensibility and specificity of the LAMP caf1 were reviewed using appropriate statistical tests. In this purpose, we replaced in the “Materials and Methods” section the paragraph “Sensitivity and specificity of LAMP caf1” to “Statistical analysis” which highlighted the method used for sensitivity and specificity estimation (see page 11, line 206 of the revised manuscript).

The issues about diagnostic performance should be addressed in an inteligent manner

Answer: Following these requirements from the Academic editor and reviewers, statements on the diagnostic performance were added in the manuscript for better clarification.

All issues and points raised by the reviewers must be addressed

Answer: Thank you very much. We have answered to all the issues and questions raised by the reviewers. See our Answers below.

The figures need to be of better quality

Answer: Thank you. We also reviewed all the figures to meet the journal’s requirements for quality.

Journal requirements

Answer: The revised manuscript meets PLOS ONE's style requirements 

Answer: Original underlying images for all gel data reported in the revised manuscript are provided as Supporting Information file (pdf file) with specific details.

Answer: Gel image data in Supporting Information was mentioned in the revised cover letter 

3. We note you have two different tables labeled as Table 3. Please ensure that all your tables are numbered separately.

Answer: The numbers of tables were corrected accordingly:

Page 11, line 199 (previous version) and Page 11, line 205 (revised version): Table 3. “Classification of the LAMP results” was deleted. 

The paragraph entitled “Sensitivity and specificity of LAMP caf1” has been renamed to “Statistical analysis” to highlight the statistical method used to assess the performance of LAMP caf1.

Page 15, line 272 (previous version) and Page 15, line 279 (revised version): Table 3. now refers to “Evaluation of LAMP caf1 performance” 

Review Comments to the Authors

Reviewer #1: The manuscript by Lovasoa Nomena Randriantseheno describes the development of a LAMP assay that would be able to detect Y. pestis in human biological samples, and thereby improving the diagnosis of plague and circumventing cumbersome bacteriological culture.

1- The authors claim that no ethical approval is needed for his study and that there is no informed consent needed from study cases. Although I believe that this might be right in this specific case a document supporting this would be good to have.

Answer: Thank you for asking for this supporting document. Although not available online, the French hard copy version of the Plague National Control Program (Edition 2012) of the Malagasy Ministry of Public Health stated at page 7, paragraph 4.4 that the notification of each plague case is mandatory at national level. It consists in recording all the necessary and useful information concerning each suspect plague patient as well as the epidemiological context in the official individual declaration form, a copy of which will be sent with the sample to the central laboratory for plague for confirmation, the surveillance of the sensitivity of strains to antibiotics and for the international weekly declaration.

We would also want to notice that these statements were already mentioned as such in other publications (see list below) using biological samples and/ or Y. pestis strains derived from plague suspected patients of Madagascar, see below: 

- Vogler AJ, Andrianaivoarimanana V, Telfer S, Hall CM, Sahl JW, Hepp CM, Centner H, Andersen G, Birdsell DN, Rahalison L, Nottingham R, Keim P, Wagner DM, Rajerison. M. Temporal phylogeography of Yersinia pestis in Madagascar: Insights into the long-term maintenance of plague. PLoS Negl Trop Dis. 2017 Sep 5;11(9):e0005887. doi: 10.1371/journal.pntd.0005887. eCollection 2017 Sep. 

- Vogler AJ, Chan F, Wagner DM, Roumagnac P, Lee J, Nera R, Eppinger M, Ravel J, Rahalison L, Rasoamanana BW, Beckstrom-Sternberg SM, Achtman M, Chanteau S, Keim P. Phylogeography and molecular epidemiology of Yersinia pestis in Madagascar. PLoS Negl Trop Dis. 2011 Sep;5(9):e1319. doi: 10.1371/journal.pntd.0001319. Epub 2011 Sep 13. 

2- The work aimed to develop a rapid, simple and sensitive/specific LAMP method for the detection of the caf1 gene sequence that is specific to Y. pestis and to evaluate its performance on biological samples from plague suspected patients from Madagascar. However, apparently previously a LAMP assay has been developed see line 100. The authors should specify why they undertook the current study. What is their novel approach.

Answer: References cited at line 100-101 are related to the development of Yersinia pestis LAMP assay using pure cultures of Y. pestis (ref#20: de Lira Nunes M et al., 2014) and simulated samples (ref#21: Feng N et al, 2017) which are already clearly stated at the introduction section. These LAMP assays were not evaluated on biological samples collected from plague suspected patients which was the aim of our study. Therefore we think that our LAMP technique was the first which was evaluated on true plague samples as they are intended to be used later.

3- The development of the LAMP assay is straightforward and well described.

Diagnostic characteristics of the test would have been more convincing if also patient samples of cases with other diseases were tested. Can the authors do this?

Answer: Thank you for this interesting suggestion. We were able to add 47 patient samples with other diseases to test on our LAMP caf1. They are distributed as follows: schistosomiasis (n = 7), taeniasis (n = 3), cysticercosis (n = 8), filariosis (n = 1) and tuberculosis (n = 28). They were kindly provided by other research units from the Institut Pasteur de Madagascar from their samples collection.

They did not show cross reactions.

4- Sensitivity and specificity should be presented with 95% confidence intervals (also in the abstract). Agreement between tests (kappa values) must be estimated and presented. In particular there should be a clear definition of the reference test. This is now presented partly in the discussion but must be included in the Methods section and presented in the Results section.

Answer: Thank you for pointing this out. The presentation of Sensitivity and specificity of the test was reviewed according to the requirement of the reviewers i.e. using 95% confidence intervals. Agreement between tests (kappa values) were also estimated and presented in all the section of the manuscript (Abstract, Materials and Methods, Results and Discussion).

We used bacteriology with isolation of Y. pestis strain as reference test for plague diagnosis which is the gold standard as per the WHO (Wer, 2006)

5- Figures are not of very good quality. This must be improved.

Answer: We agree with the reviewer’s comment. They were improved to meet the journal requirement accordingly.

Reviewer #2: The manuscript addresses the development of a loop-mediated isothermal amplification (LAMP) test to aid in the diagnosis of Plague by detecting Yersinia pestis nucleic acids in clinical samples.

The authors explain the advantages of polymerase chain reaction (PCR) in the diagnosis of Plague, in opposition to culture and antigen detection rapid diagnostic test (RDT). The aim of the study is to develop a LAMP method that being simpler than PCR can replace it and implemented at lower levels of the health system, provided it has good diagnostic performance.

The methodology used is appropriate, but I would like to discuss some aspects of this work:

1) According to the authors PCR alone is not enough to confirm a diagnosis of Plague and this requires also a positive RDT result or a 4X increase in antibody titre, why is this required? Please, explain. Why not aiming at a LAMP method that can be used alone as confirmatory tool?

Answer: We refer to the plague case definition according to the WHO (see WER, 2006) which stated that a confirmed plague case requires: a culture positive with isolation of Y. pestis OR a combination of PCR and RDT positive OR a 4X increase in antibody titre. To date, there is no updated guideline from the WHO which can validate the use of PCR alone as a confirmatory test. However, as it was designed to be used at least at the district hospital, at the patient’s bedside, the plague LAMP method can help physicians (working at the health centers in remote plague areas) to confirm plague cases when combined with F1 RDT. Indeed, F1 RDT is already widely used in plague endemic areas since 2002.

2) The LAMP method proposed is home brewed, will this be accepted by the Ministry of Health? Is the PCR method also home brewed?

Answer: We think that this will be accepted by the MoH since almost biological tools used at the CLP for plague diagnosis followed home-made protocols. In addition, they were all validated on biological samples as required by the WHO.

3) When cross-reactions were assessed was DNA preparation and measurement the same for all samples? This is not clear.

Answer: Fourteen pathogens were tested for cross-reactions with LAMP caf1. Most of these pathogens were cultured in CLP and DNA extraction was performed using DNeasy Blood & Tissue Kits. However, for Taenia solium, Plasmodium vivax, Plasmodium falciparum and Mycobacterium tuberculosis, DNA extraction was conducted by other research units of the Institut Pasteur de Madagascar following their own protocols. The methods may be slightly different compared to our extraction protocol: DNeasy Blood & Tissue Kit (Qiagen Kit) used for Taenia solium, Plasmodium vivax, Plasmodium falciparum and an in-house protocol for Mycobacterium tuberculosis. 

However, DNA measurements of all these pathogens were performed using Nanodrop 2000 but no DNA concentration adjustment was done before testing. Therefore, we added clarifications in the manuscript according to the reviewer’s comment.

4) LAMP shows lower sensitivity than culture. Is not this a suboptimal performance? 

Answer: It is true that compared to culture, LAMP caf1 shows a lower sensitivity. However, in most of countries affected by plague, having additional test which can be used at the peripheral level allows health care workers to confidently rule in cases mainly for pneumonic plague, a severe and contagious disease, which may be clinically confused with acute respiratory infection (IRA).

Also, the use of SYBRGreen is a big limitation, the authors suggest to quantify DNA first and adjust DNA concentration, which is not ideal for a point-of-care (POC) test. Also, for coloured DNA samples analysis of amplified products by PCR is also suggested, which again is far from a POC.

Answer: We actually highlighted some problems related to the use of Sybr Green I during this study. While testing M. tuberculosis DNA for cross-reactions, we found that high concentration of DNA can lead to a false positive result with Sybr Green I. We proposed to quantify the DNA extract before amplification but for point-of-care diagnostic, we understood that this is not the ideal scenario. That is why we proposed an alternative method consisting in adding Sybr Green I to an aliquot of DNA before starting the amplification reaction. This is to make sure that the DNA concentration of the pre-treated sample is not too high and will not lead to a false positive result, if Sybr Green I turns green with this prior test, a simple dilution of the pre-treated sample is needed before starting the amplification. Regarding the dark colored sample, of the 113 samples tested, only one sample was dark colored and led to misinterpretation of the colorimetric result. For this sample, we run LAMP amplicons in a gel electrophoresis and saw the ladder-like pattern to classify it as a positive reaction. This case is an exception. Indeed, the biological samples collected for plague tests are either bubo aspirates or sputum samples which are usually not dark colored. For the case of septicemic plague, which is rarely reported, blood samples are collected and additional steps in the pre-treatment procedure can be applied to avoid such problems as reported previously (Viana GMR, Silva-Flannery L, Lima Barbosa DR, et al. Field evaluation of a real time loop-mediated isothermal amplification assay (RealAmp) for malaria diagnosis in Cruzeiro do Sul, Acre, Brazil. PLoS One. 2018;13(7):e0200492. Published 2018 Jul 11. doi:10.1371/journal.pone.0200492). However, no blood samples were included in our study.

5) Sensitivity and specificity are not estimated with confidence intervals, please apply the statistics needed for this.

Answer: Thank you for this remark. Sensitivity and specificity are now estimated with 95% confidence intervals as recommended by the reviewers. Agreement between tests (kappa values) were also estimated and presented in the whole manuscript.

6) The resulting LAMP test, as it is, does not seem to bring any advantage to the diagnosis of Plague

Answer: We don’t agree with the reviewer’s comment. According to the performance of the LAMP test developped in this study (Sensitivity of 97.9% and Specificity of 94.6% compared to the gold standard which is the bacteriological culture) and its practicability particularly for low resource countries such as Madagascar, it is a promising technology for plague diagnosis. Indeed, low income countries where such infectious disease is still endemic have a urgent need to develop diagnostic tool which can be performed at the level of the district hospital without the use of sophisticated equipment and devices. In addition to the F1RDT (already used at the level of plague endemic districts), the implementation of LAMP PCR will allow the confirmation of plague cases on site. This will significantly reduce the time for confirmation and therefore the implementation of appropriate preventive measures to stop the transmission.

---

## [Decision Letter · Decision Letter 1]

29 May 2020

PONE-D-20-06493R1

Development and evaluation of loop-mediated isothermal amplification for detection of Yersinia pestis in plague biological samples

PLOS ONE

Dear Dr. Andrianaivoarimanana,

Thank you for submitting your manuscript to PLOS ONE. After careful consideration, we feel that it has merit but does not fully meet PLOS ONE’s publication criteria as it currently stands. Therefore, we invite you to submit a revised version of the manuscript that addresses the points raised during the review process.

You are requested to address the issues raised by the reviewers. In particular:

- provide statement/evidence for ethical approval;

- consult native English speaking person to improve use of English

- try to improve the quality of the figures (high resolution)

Also ensure that you follow PLoS ONE policy: The revised submission should include the raw blot/gel image data for your review, either in Supporting Information or via a public data repository; the Data Availability Statement should indicate where these data can be found. The original blot/gel image data should (1) represent unadjusted, uncropped images, (2) be provided for all blot/gel data reported in the main figures and Supporting Information, and (3) match the images in the manuscript figure(s).

We look forward to receiving your revised manuscript.

Kind regards,

Henk D. F. H. Schallig, Ph.D

Academic Editor

PLOS ONE

Reviewers' comments:

Reviewer's Responses to Questions

**Comments to the Author**

1. If the authors have adequately addressed your comments raised in a previous round of review and you feel that this manuscript is now acceptable for publication, you may indicate that here to bypass the “Comments to the Author” section, enter your conflict of interest statement in the “Confidential to Editor” section, and submit your "Accept" recommendation.

Reviewer #1: (No Response)

Reviewer #2: All comments have been addressed

2. Is the manuscript technically sound, and do the data support the conclusions?

Reviewer #1: Yes

Reviewer #2: Yes

3. Has the statistical analysis been performed appropriately and rigorously? 

Reviewer #1: Yes

Reviewer #2: Yes

4. Have the authors made all data underlying the findings in their manuscript fully available?

Reviewer #1: Yes

Reviewer #2: Yes

5. Is the manuscript presented in an intelligible fashion and written in standard English?

Reviewer #1: No

Reviewer #2: Yes

6. Review Comments to the Author

Reviewer #1: Most of my comments have been addressed and some addiitonal samples seem to be analysed. I would like to see a statement on the patients'consent that these addiitonal sample could have been used for this study (and/or ethical review outcome that supports this). Also the relevant part of the document the authors refer to that cover the ethical approval for the study should be uploaded as suplementary material

English may require some attention

Quality of figures remains a concenr

Reviewer #2: (No Response)

7. PLOS authors have the option to publish the peer review history of their article (what does this mean?). If published, this will include your full peer review and any attached files.

Reviewer #1: No

Reviewer #2: No

---

## [Author Response · Author response to Decision Letter 1]

10 Jul 2020

Review Comments to the Author

Reviewer #1: 

Most of my comments have been addressed and some addiitonal samples seem to be analysed. I would like to see a statement on the patients'consent that these addiitonal sample could have been used for this study (and/or ethical review outcome that supports this). 

Answer: Additional samples used for this study have been anonymized and aggregated precluding identification of patients at individual level. They are part of residual samples from:

- Research project authorized for further use in future research (following the reference # 041-MSANP/CERBM du 08 juin 2017 from the Ethics Committee for Biomedical Research of the Malagasy Ministry of Public Health).

- Tuberculosis confirmation: mandatory samples collection as part of Madagascar’s routine surveillance system under national public health law. 

They were all provided in the purpose of Public Health improvement in Madagascar.

Also the relevant part of the document the authors refer to that cover the ethical approval for the study should be uploaded as suplementary material

Answer: A copy of the Plague National Control Program (Edition 2012) of the Malagasy Ministry of Public Health has been uploaded as supplementary material (S1 Appendix. Plague National Control Program).

English may require some attention

Answer: The whole manuscript has now been reviewed by Dr Dawn Birdsell - a native English speaking scientist- for English improvement.

Quality of figures remains a concenr

Answer: All figure files have been greatly improved in terms of resolution to meet the Journal requirements. They have also been uploaded to the Preflight Analysis and Conversion Engine (PACE) digital diagnostic tool, https://pacev2.apexcovantage.com/ as required by the Journal.

---

## [Editor Report · Decision Letter 2]

31 Jul 2020

Development and evaluation of loop-mediated isothermal amplification for detection of Yersinia pestis in plague biological samples

PONE-D-20-06493R2

Dear Dr. Andrianaivoarimanana,

We’re pleased to inform you that your manuscript has been judged scientifically suitable for publication and will be formally accepted for publication once it meets all outstanding technical requirements.

Kind regards,

Henk D. F. H. Schallig, Ph.D

Academic Editor

PLOS ONE
---

## [Editor Report · Acceptance letter]

6 Aug 2020

PONE-D-20-06493R2 

Development and evaluation of loop-mediated isothermal amplification for detection of *Yersinia pestis* in plague biological samples 

Dear Dr. Andrianaivoarimanana:

I'm pleased to inform you that your manuscript has been deemed suitable for publication in PLOS ONE. Congratulations! Your manuscript is now with our production department. 

Kind regards, 

on behalf of

Dr. Henk D. F. H. Schallig 

Academic Editor

PLOS ONE